

# COADREADx: A comprehensive algorithmic dissection of colorectal cancer unravels salient biomarkers and actionable insights into its discrete progression

Ashok Palaniappan[1], Sangeetha Muthamilselvan[1] and Arjun Sarathi[2]

[1] Systems Computational Biology Lab, Department of Bioinformatics, School of Chemical and Biotechnology, SASTRA Deemed University, Thanjavur, Tamil Nadu, India
[2] Novo Nordisk Foundation Center for Basic Metabolic Research, Faculty of Health and Medical Sciences, University of Copenhagen, Copenhagen, Denmark

Corresponding author
Ashok Palaniappan,
apalania@scbt.sastra.edu

## ABSTRACT

**Background**. Colorectal cancer is a common condition with an uncommon burden of disease, heterogeneity in manifestation, and no definitive treatment in the advanced stages. Renewed efforts to unravel the genetic drivers of colorectal cancer progression are paramount. Early-stage detection contributes to the success of cancer therapy and increases the likelihood of a favorable prognosis. Here, we have executed a comprehensive computational workflow aimed at uncovering the discrete stagewise genomic drivers of colorectal cancer progression.

**Methods**. Using the TCGA COADREAD expression data and clinical metadata, we constructed stage-specific linear models as well as contrast models to identify stage-salient differentially expressed genes. Stage-salient differentially expressed genes with a significant monotone trend of expression across the stages were identified as progression-significant biomarkers. The stage-salient genes were benchmarked using normals-augmented dataset, and cross-referenced with existing knowledge. The candidate biomarkers were used to construct the feature space for learning an optimal model for the digital screening of early-stage colorectal cancers. The candidate biomarkers were also examined for constructing a prognostic model based on survival analysis.

**Results**. Among the biomarkers identified are: CRLF1, CALB2, STAC2, UCHL1, KCNG1 (stage-I salient), KLHL34, LPHN3, GREM2, ADCY5, PLAC2, DMRT3 (stage-II salient), PIGR, HABP2, SLC26A9 (stage-III salient), GABRD, DKK1, DLX3, CST6, HOTAIR (stage-IV salient), and CDH3, KRT80, AADACL2, OTOP2, FAM135B, HSP90AB1 (top linear model genes). In particular the study yielded 31 genes that are progression-significant such as ESM1, DKK1, SPDYC, IGFBP1, BIRC7, NKD1, CXCL13, VGLL1, PLAC1, SPERT, UPK2, and interestingly three members of the LY6G6 family. Significant monotonic linear model genes included HIGD1A, ACADS, PEX26, and SPIB. A feature space of just seven biomarkers, namely ESM1, DHRS7C, OTOP3, AADACL2, LPHN3, GABRD, and LPAR1, was sufficient to optimize a RandomForest model that achieved > 98% balanced accuracy (and performant recall) of cancer vs. normal on external validation. Design of an optimal multivariate model

based on survival analysis yielded a prognostic panel of three stage-IV salient genes, namely HOTAIR, GABRD, and DKK1. Based on the above sparse signatures, we have developed COADREADx, a web-server for potentially assisting colorectal cancer screening and patient risk stratification. COADREADx provides uncertainty measures for its predictions and needs clinical validation. It has been deployed for experimental non-commercial use at: https://apalanialab.shinyapps.io/coadreadx/.

## INTRODUCTION

Colorectal adenocarcinoma (COADREAD), or colorectal cancer, is a common cancer with about 1.9 million cases and 930,000 deaths occurring in 2020 (*Morgan et al., 2023*). There are many lifestyle and environmental drivers of colorectal cancer apart from family history, making the bulk of its incidence sporadic (*Haggar & Boushey, 2009*). Some of these drivers include dietary concerns (*Willett, 2005*), physical inactivity, obesity (*de Jong et al., 2005*), alcohol and tobacco (*Zisman et al., 2006*), *etc.* Familial forms of colorectal cancer include (i) familial adenomatous polyposis (FAP) associated with mutations in the APC tumor suppressor gene (TSG) (*Wilmink, 1997*); and (ii) hereditary nonpolyposis colorectal cancer (HNPCC, Lynch syndrome) associated with mutations in the DNA repair pathway genes, MSH2 and MLH1 (*Haggar & Boushey, 2009*). Since survival rates in colorectal cancer plummet with late-stage of presentation, effective surveillance *via* access to screening models is necessary. Early-stage diagnosis of colorectal cancer is essential to secure an advantageous prognosis, which could help in the clinical management of the disease.

The Cancer Genome Atlas (TCGA) research network has found mutational and integrative signatures in the multidimensional COADREAD dataset (*The Cancer Genome Atlas Network, 2012*), but so far our knowledge with respect to the stage-wise progression of colorectal cancer has been incomplete and inadequate. It is known that gene expression profiles of certain markers define cell-type identity (*Chen et al., 2018*), and even tissue microenvironment (*Luca et al., 2021*), it is reasonable to suppose that a community structure of cell-types drives colorectal cancer progression. Molecular gene signatures characterize the cell composition of the tumor, and it could be argued that the tumor progression through stages is in part or whole determined by the complex and collective changes in gene expression. The AJCC staging of colorectal cancer is based on histopathology (*viz.* the TNM staging) (*Amin et al., 2017*), and it would be interesting to study the evidence for a molecular basis of cancer progression in discrete stages.

We developed data-driven workflows for discerning the molecular signatures of colorectal cancer through RNA-Seq transcriptomics. We extended the protocol introduced in *Sarathi & Palaniappan (2019)*, and identified stage-salient biomarkers. A new class of

biomarkers with a significant monotone trend of differential expression, called progression-significant DEGs, were also identified. It is noted that the early-stage (*i.e.*, stage-I and stage-II salient) biomarkers could be useful in development of diagnostics and prognostic models, whereas progression-significant biomarkers could pinpoint potential therapeutic targets to halt or reverse the course of cancer (before it does metastasize to a point of no return). A network analysis grounds the findings in a larger context, lending more evidence for the molecular origins of stage-wise discrete cancer progression. Based on the above results, we have developed models for the early-stage screening as well as risk stratification of colorectal cancer. These models were bundled into COADREADx, a pilot tool for the digital diagnostic and prognostic screening of colorectal cancers. COADREADx is available at: https://apalanialab.shinyapps.io/coadreadx/ as a user-friendly interface for academic use. Source code is available from: https://zenodo.org/doi/10.5281/zenodo.13790219. All original datasets used in the study were obtained from the public-domain, and all the intermediate results generated from the study are available as Supplementary Information (DOI: 10.6084/m9.figshare.20489211.v5). Portions of this text were previously published as part of a preprint (https://www.medrxiv.org/content/10.1101/2022.08.16.22278877v3).

# MATERIAL AND METHODS

The workflow is summarized in Fig. 1 and discussed in detail below. The identification of stage-salient biomarkers follows the computational protocol developed earlier in our lab (*Sarathi & Palaniappan, 2019*).

## Data preprocessing

Normalized and $\log_2$-transformed Illumina HiSeq RNA-Seq gene expression data for Colorectal Adenocarcinoma (COADREAD) processed by the RSEM pipeline (*Li & Dewey, 2011*) were obtained from TCGA *via* the http://firebrowse.org/ portal (accessed 06-01-2019) (*Broad Institute TCGA Genome Data Analysis Center, 2016*). The patient barcode (uuid) of each sample encoded in the variable called 'Hybridization REF' was parsed and used to annotate the controls and cancer samples. To annotate the stage information of the cancer samples, we obtained the corresponding clinical dataset from http://firebrowse.org/ and merged the clinical data with the expression data by matching the "Hybridization REF" in the expression data with the aliquot barcode identifier in the clinical data. The cancer staging is encoded in the attribute "pathologic_stage" of the clinical data. The sub-stages (A,B,C) were collapsed into the parent stage, resulting in four stages of interest (I, II, III, IV). We retained a handful of clinical variables related to demographic features, namely age, sex, height, weight, and vital status. Using this merged dataset, we filtered out genes that showed little change in expression across all samples (defined as $\sigma < 1$). We also removed cancer samples that were missing stage annotation (value 'NA' in the "pathologic stage") from our analysis. Data pre-processing was done with R v4.2.3 (www.r-project.org) and the final dataset was processed through voom function in R limma v3.54.2 to prepare for linear modeling (*Law et al., 2014*).
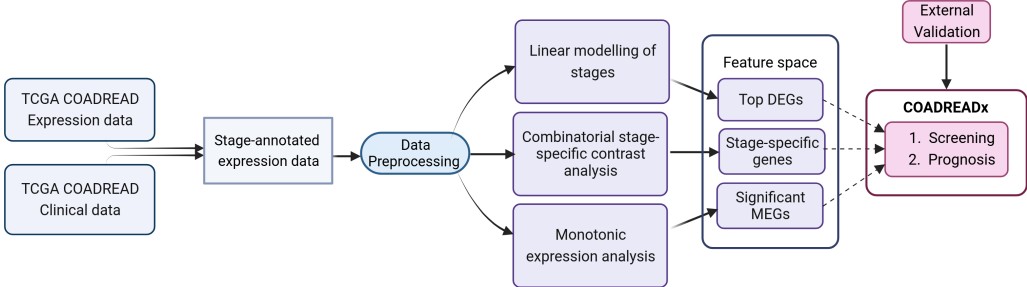

**Figure 1** **Study design for the dissection of discrete stage-wise progression of colorectal cancer.** The identified candidate biomarkers could be used to train machine learning classifiers for the screening and prognosis of colorectal cancers. Figure created with Biorender.com.

## Linear modelling

Linear modeling of expression across cancer stages relative to the baseline expression (*i.e.,* in normal tissue controls) was performed for each gene using R limma v3.54.2 (*Ritchie et al., 2015*). The following linear model was fit for each gene's expression based on the design matrix shown in Fig. 2A:

$$y = \alpha + \beta_1 x_1 + \beta_2 x_2 + \beta_3 x_3 + \beta_4 x_4 \tag{1}$$

where the independent variables are indicator variables of the sample's stage, the intercept $\alpha$ is the baseline expression estimated from the controls, and $\beta_i$ are the estimated stagewise log fold-change (lfc) coefficients relative to controls. The linear model was subjected to empirical Bayes adjustment to obtain moderated t-statistics (*McCarthy & Smyth, 2009*). To account for multiple hypothesis testing and the false discovery rate, the *p*-values of the F-statistic of the linear fit were adjusted using the method of *Hochberg & Benjamini (1990)*. The linear trends across cancer stages for the top significant genes were visualized using boxplots to ascertain the regulation status of the gene relative to the control.

## Pairwise contrasts

To perform contrasts, a slightly modified design matrix shown in Fig. 2B was used, which would give rise to the following linear model of expression for each gene:

$$y = \beta_0 x_0 + \beta_1 x_1 + \beta_2 x_2 + \beta_3 x_3 + \beta_4 x_4 \tag{2}$$

where the controls themselves constitute one of the indicator variables, and the $\beta_i$ are all coefficients estimated only from the corresponding samples. Our first contrast of interest, between each stage and the control, was achieved using the contrast matrix shown in Table 1. Four contrasts were obtained, one for each stage *vs* control. A threshold of |lfc|>2 was applied to each contrast to identify genes differentially expressed with respect to the control. Genes could be differentially expressed in any combination of the stages. In the first pass, we identified genes whose |lfc|>2 for any stage. For the genes that passed, we identified the stage that showed the highest |lfc| for each gene and assigned the gene as specific to that stage for the rest of our analysis.

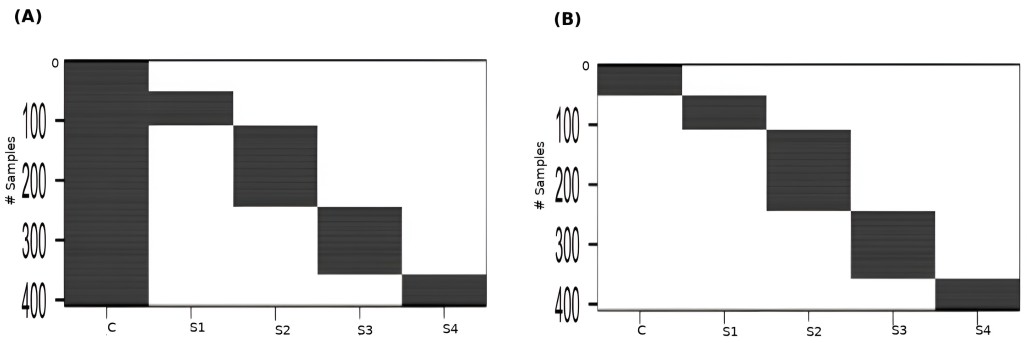

**Figure 2** Design matrices used for (A) linear modeling ; and (B) between-stages contrasts. C: Control, S1: Stage-I, S2: Stage-II, S3: Stage-III, S4: Stage-IV.

**Table 1** Coefficients of the contrasts matrix for stage-control modeling of the expression matrix.

| Clinical annotation | STAGE - CONTROL CONTRASTS | | | |
|---|---|---|---|---|
| | I | II | III | IV |
| Control | −1 | −1 | −1 | −1 |
| Stage1 | 1 | 0 | 0 | 0 |
| Stage2 | 0 | 1 | 0 | 0 |
| Stage3 | 0 | 0 | 1 | 0 |
| Stage4 | 0 | 0 | 0 | 1 |

**Table 2** Coefficients of the contrasts matrix for between-stages modelling of the annotated expression matrix.

| Clinical annotation | CONTRAST BETWEEN STAGES | | | | | |
|---|---|---|---|---|---|---|
| | (I, II) | (I, III | (II, III) | (I, IV) | (II, IV) | (III, IV) |
| Control | 0 | 0 | 0 | 0 | 0 | 0 |
| Stage1 | −1 | −1 | 0 | −1 | 0 | 0 |
| Stage2 | 1 | 0 | −1 | 0 | −1 | 0 |
| Stage3 | 0 | 1 | 1 | 0 | 0 | −1 |
| Stage4 | 0 | 0 | 0 | 1 | 1 | 1 |

## Significance analysis

We applied four-pronged criteria to establish the salience of the stage-specific differentially expressed genes. (i) Adj. $p$-value of the contrast with respect to the control $< 0.001$ (ii)–(iv) $P$-value of the contrast with respect to other stages $< 0.05$. Six such contrasts are possible.

To obtain the above $p$-values (ii)–(iv), we used the contrast matrix shown in Table 2, which was supplied as an argument to the contrastsFit function in *limma*.

To deal with any sparsity of progression-significant genes salient to any stage, we defined the "pval_pdt" of a given gene in a certain stage as the product of the p_values of its expression contrast in that stage *vs* each of the other stages (*e.g.*, pval_pdt of gene x in stage 1 is (pval(gene x in st1 *vs* st2))*(pval(gene x in st1 *vs* st3))*(pval(gene x in st1 *vs* st4)).

## Monotonic expression

The linear model in Eq. (1) would not be sufficient to identify genes with an monotonic, trend of expression in sync with disease progression, which could uncover stage-agnostic expression of progression-significant driver genes. Towards this end, we used a model of gene expression where the cancer stage was treated as a numeric variable:

$$y = aX + b \tag{3}$$

where X takes a value in (0,1,2,3,4) corresponding to the sample stage: (control, I, II, III, IV), respectively. It was noted the mean gene expression could show the following patterns of monotonic expression across cancer stages:

(i) monotonic upregulation, where mean expression follows:
control <I <II <III <IV.
(ii) monotonic downregulation, where mean expression follows:
control >I >II >III >IV.

The sets of genes conforming to either (i) or (ii) were identified to yield monotonically upregulated and monotonically downregulated genes. These two sets were merged, and the final set of genes was evaluated using the adj. *p*-values from the model given by Eq. (3) to yield genes with significant monotonic patterns of expression.

## Models for cancer screening and prognosis

### Validation of biomarkers with normals-augmented dataset

To study the reliability of findings when a reasonable number of controls are used, we augmented the TCGA cohort with the COADREAD dataset from RNAseqDB (*Wang et al., 2018*) (accessed 12-06-2022) that couples TCGA data with 339 normals from the Genotype-Tissue Expression (GTEx) database (*GTEx Consortium, 2013*). The consolidated dataset was subjected to the same biomarker protocol to identify stage-salient genes, and the results compared with those obtained with the TCGA dataset.

### Development of diagnostic model

The different classes of biomarkers discussed above, including stage-salient genes and monotonically expressed genes, could be used as the feature space to train machine learning (ML) algorithms to solve the binary classification problem of cancer v/s normal samples (*Muthamilselvan, Ramasami Sundhar Baabu & Palaniappan, 2023*). Towards this, we split the TCGA dataset in the ratio 0.8:0.2 stratified on the outcome class ('cancer' or 'normal'), and extracted the features of interest. To reduce the dimensionality of the feature space, feature selection techniques such as R Boruta v8.0.0 (*Kursa & Rudnicki, 2010*) and recursive feature elimination (in R caret v6.0.94 *Kuhn, 2008*) were applied to the train dataset and a consensus reduced feature space was obtained. Different ML algorithms were trained on this feature space and hyperparameters optimized by cross-validation. The performance of the ML algorithms was evaluated on the holdout testset to determine the best ML model. The best-performing ML model was then validated on external out-of-domain cohorts.

### Development of prognostic model

To study the prognostic significance of the identified stage-salient genes, we used the patient 'OS' ('Overall Survival') attribute in the clinical metadata of the TCGA cohort

(accessed 06-01-2019). Survival analysis was performed according to the protocol outlined in *Muthamilselvan & Palaniappan (2023)*. Univariate Cox regression analysis of the top stage-salient genes was executed to screen the prognostically significant ones, using the R survival library v3.5.7 (*Therneau & Lumley, 2015*). Genes with $p$-value $< 0.05$ were regarded as candidate genes for building a multivariate Cox regression model. This was done using backward variable selection based on the model's Akaike Information Criterion (AIC) metric (*Gerds, Scheike & Andersen, 2012*). The procedure yielded an optimal prognostic signature of size n, given by the following equation:

$$Risk\ score = \beta_1 * gene_1 + \cdots + \beta_i * gene_i + \cdots + \beta_n * gene_n \qquad (4)$$

where the $\beta_i$ are the coefficients for the expression of the ith gene. The median risk score from the above distribution was used to classify TCGA COADREAD patients into high-risk and low-risk groups, as implemented in R survminer library v0.4.9 (*Kassambara et al., 2017*). Kaplan–Meier analysis was then performed to assess significance in survival rate variations between the high-risk and low-risk groups, and thereby qualify the biomarker signature.

## Benchmarking

Principal component analysis (PCA) was performed using prcomp in R. We used the rand function to choose 100 random genes. In order to visualize significant outlier genes with a large effect size, volcano plots were obtained by plotting the ($-\log_{10}$)-transformed $p$-value *vs.* the log fold-change of gene expression. Heat maps of significant stage-salient differentially expressed genes were visualized using R pheatmap v1.0.12 and clustered using R hclust function. Novelty of the identified stage-salient genes was ascertained by screening against curated databases, including the Cancer Gene Census (CGC at https://cancer.sanger.ac.uk/cosmic; accessed 01-12-23) (*Futreal et al., 2004*), Network of Cancer Genes NCG7.0 (accessed 01-12-23) (*Repana et al., 2019*), and the Clinical Trials Registry (http://www.clinicaltrials.gov; accessed 01-12-23). STRINGdb was used to translate the findings into network-level insights (*Szklarczyk et al., 2021*). To perform immuno-cyte infiltration analysis, we used Cibersort and estimated the proportion of tumor-infiltrating immune cells in TCGA COADREAD samples based on gene expression signatures (*Amin et al., 2017*; *Newman et al., 2019*). Cibersort's inbuilt LM22 signature estimated the proportion of 22 standard immune cell types; setting the number of permutations to 100 allowed the calculation of sample-wise statistical significance with respect to the estimated values. The immuno-cyte patterns of significant samples were analyzed to provide a snapshot of immune ecotypes at play in significant tumor and normal samples, which would increase our basic understanding of colorectal cancer pathologies and advance rational therapies. The cell-type correlation matrix computed from the proportions of cell-types across significant samples was used to identify substantial co-occurrence patterns. The relative abundance of immunocytes between tumor and normal samples was compared to pinpoint significant differentially elevated or depressed tumor-infiltrating immune cells.
**Table 3 Stagewise distribution of colorectal cancer samples.**

| TCGA Stage | TNM classification | # Cases | |
|---|---|---|---|
| 1 | T1a N0 M0 | 56 | 57 |
| 1A | T1b N0 M0 | 1 | |
| 2 | T2 N0 M0 | 18 | |
| 2A | T2 N0 M0 | 110 | 136 |
| 2B | T2 N0 M0 | 6 | |
| 2C | T2 N0 M0 | 2 | |
| 3 | T3 N0 M0 | 9 | |
| 3A | T4 N0 M0 | 10 | 113 |
| 3B | – | 59 | |
| 3C | – | 35 | |
| 4 | – | 27 | |
| 4A | T(any) N1 M0 | 23 | 52 |
| 4B | T(any) N(any) M1 | 2 | |
| CONTROL | – | 51 | |
| NA | – | 19 | |

**Table 4 Statistical summary of clinical meta-data associated with the TCGA COADREAD transcriptome.** Numeric attributes are presented as mean ± standard deviation. Nominal attributes (gender and vital status) are presented as counts. BMI could be calculated for patients with both height and weight data.

| Characteristic | | Control | STAGE OF CRC | | | | NA | Overall |
|---|---|---|---|---|---|---|---|---|
| | | | I | II | III | IV | | |
| Number of samples | | 51 | 57 | 136 | 113 | 52 | 19 | 428 |
| Age (years) | | 69.1 ± 14.1 | 65.8 ± 12.6 | 66.7 ± 12.9 | 63.1 ± 13.2 | 60.6 ± 13.3 | 65.4 ± 12.2 | 65.1 ± 13.3 |
| Weight (kg) | | 79.3 ± 25.3 | 83.9 ± 19.4 | 78.3 ± 23.3 | 81.3 ± 20.2 | 82.2 ± 17.4 | 83.6 ± 26.2 | 80.7 ± 21.5 |
| Height (cm) | | 169.3 ± 9.5 | 172.1 ± 11.0 | 167.0 ± 13.0 | 169.0 ± 11.0 | 172.0 ± 11.1 | 170.9 ± 12.3 | 169.2 ± 11.7 |
| BMI (kg/m$^2$) | | 27.4 ± 7.0 | 28.5 ± 6.1 | 29.7 ± 25.1 | 28.3 ± 6.3 | 28.7 ± 5.6 | 28.1 ± 6.0 | 28.8 ± 15.3 |
| Gender | Male | 23 | 34 | 72 | 61 | 30 | 11 | 231 |
| | Female | 28 | 23 | 64 | 52 | 22 | 8 | 197 |
| Vital status | Alive | 44 | 55 | 122 | 100 | 36 | 15 | 372 |
| | Dead | 7 | 2 | 14 | 13 | 16 | 4 | 56 |

## RESULTS

The gene expression matrix from TCGA consisted of 20,502 genes × 428 samples. Upon data pre-processing, the gene expression matrix consisted of 18,212 genes × 409 samples, with an additional vector denoting the sample stage. This dataset is made available as File S1. Table 3 shows the distribution of TCGA samples with the corresponding AJCC staging. Table 4 shows a summary of patient demographic characteristics, with mean age ∼65 years and mean body mass index (BMI) ∼29 hinting at etiological roles of ageing and obesity.

After preprocessing with voom in *limma,* (*Law et al., 2014*), the dataset yielded 9,433 significant genes (adj. P <1E-5) in the linear modeling, suggesting the existence of a linear

Palaniappan et al. (2024), *PeerJ*, DOI 10.7717/peerj.18347    8/35

**Table 5 Stage-wise lfc, and inferred regulation status of the top ten genes from the linear modelling analysis, ranked by adjusted p-value of the linear model.** A mixture of both upregulated and downregulated genes was obtained, shown separately here.

| Gene | Stage I lfc ($\beta_1$) | Stage II lfc ($\beta_2$) | Stage III lfc ($\beta_3$) | Stage IV lfc ($\beta_4$) | Adj. p-val | Regulation status |
|------|------|------|------|------|------|------|
| CDH3 | 6.5572 | 6.4729 | 6.4325 | 6.4874 | 1.06E−156 | UP |
| KRT80 | 6.8613 | 6.6695 | 6.9847 | 7.2830 | 4.39E−143 | UP |
| ETV4 | 5.6165 | 5.5937 | 5.5175 | 5.8992 | 8.28E−131 | UP |
| ESM1 | 5.7276 | 5.9611 | 5.9339 | 6.4049 | 2.56E−130 | UP |
| JUB | 3.1785 | 3.1473 | 3.1536 | 3.0750 | 7.78E−102 | UP |
| MTHFD1L | 2.6099 | 2.5692 | 2.5300 | 2.5766 | 2.10E−100 | UP |
| OTOP2 | −9.9507 | −10.030 | −9.9761 | −9.9196 | 4.62E−139 | DOWN |
| AADACL2 | −3.3481 | −3.4103 | −3.3285 | −3.3960 | 4.99E−131 | DOWN |
| DHRS7C | −3.4279 | −3.5170 | −3.5209 | −3.5196 | 3.14E−130 | DOWN |
| OTOP3 | −5.3795 | −5.2544 | −5.1438 | −5.1531 | 1.80E−125 | DOWN |

trend in their expression across cancer stages. Such an observation could be explained by cancer hallmarks that typically worsen with progression, for *e.g.*, genome-wide instabilitya cancer hallmark, *Hanahan & Weinberg (2011)*. Some top-ranked upregulated genes from the linear modeling included CDH3, KRT80, ETV4 and ESM1. CDH13 was notably a top upregulated gene obtained from the linear modeling of hepatocellular carcinoma (only after GABRD and PLVAP) in an earlier analysis (*Sarathi & Palaniappan, 2019*); these observations point to a consistent role for members of the cadherin gene family in cancer progression in gastrointestinal cancers. The top downregulated genes included OTOP2, OTOP3, AADACL2 and DHRS7C. Table 5 shows the log-fold changes of the top ten genes in with respect to normal samples,and Boxplots of the expression of the top 9 genes indicated a progressive net increase in expression across cancer stages relative to control for up-regulated genes, while repressed expression across cancer stages relative to control was the hallmark of downregulated genes (Fig. 3). A constant trend of regulation across stages underscores the stage-specific basis of cancer progression. It is noted that the linear trend identified needs to be validated with a model for monotonic expression (see Methods), and some stage-specific genes might exhibit maximal differential expression in stages other than stage 4 (Fig. 4).

The samples were visualized using a PCA of the top 100 genes from the linear model (Fig. 5A). Separate and distinct clusters of the controls and cancer samples suggested considerable changes in gene expression in cancer samples. In contrast, the PCA plot of randomly sampled 100 genes (Fig. 5B) failed to distinguish the cancer and control samples, highlighting the potential of stagewise linear models in identifying cancer-specific genes (File S2).

Differences in gene expression constitute the basis of cell-type identities, and it may not be surprising that differences in gene expression drive cancer progression through the AJCC stages. In the first pass, we eliminated 15,970 genes with |lfc|<2 in all stages (Table 1). We binned the remaining genes into different partitions, to obtain stage-specific genes of varying sizes (Fig. 6). To establish salience, we applied the second contrast (Table 3) and checked for filter criteria (ii)–(iv) stated in the Methods section. Genes that passed

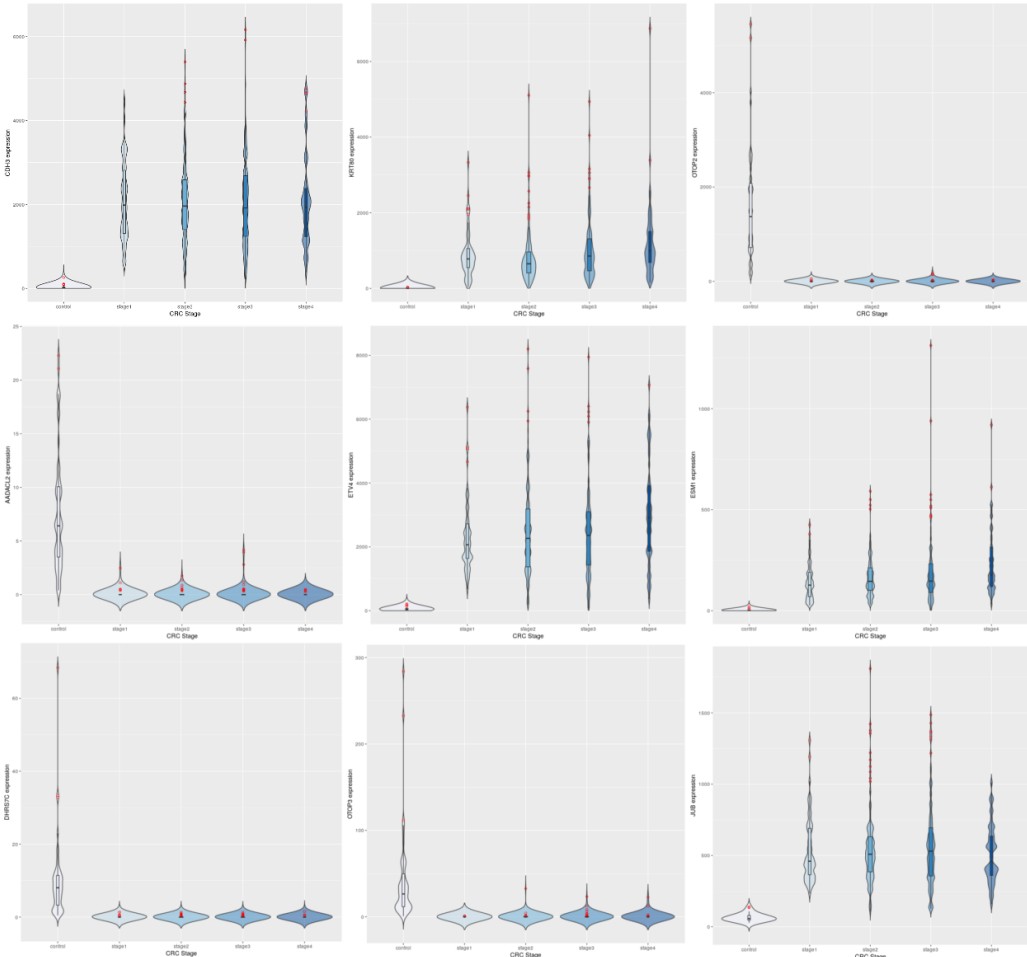

**Figure 3  Expression trends of the top 9 DEGs from the linear modeling.** Row-1: CDH3, KRT80, OTOP2; Row-2: AADACL2, ETV4, ESM1; Row-3: DHRS7C, OTOP3, JUB. In each plot, expression trends in the control samples are shown first, followed by stage-wise trends in progressive fashion. It can be observed that some genes are downregulated to near-zero expression as CRC progresses (notably OTOP2, OTOP3, AADACL2 and DHRS7C).

all filters were identified as stage-salient DEGs. This process yielded 71 stage-I salient, 2 stage-II salient, 0 stage-III salient and 59 stage-IV salient genes (File S3).

Considering the sparsity of genes passing the filters for stages 2 and 3, we applied the pval_pdt, described in the Methods section, and extracted the top 10 genes for each stage. For stages 1 and 4, all these top 10 genes figured in the 71 and 59 genes that had been identified as stage-salient DEGs, respectively. For stage 2, we took the two genes that passed the filtering and appended genes with the lowest pval_pdt to obtain 10 genes. For stage 3, we used the 17 genes with pval_pdt $< 0.125E{-}3$. The top 10 genes from each stage are shown in Table 6, and the entire set of 157 stage-salient DEGs are presented in File S3. It is significant that GABRD emerges as a stage-IV salient gene in COADREAD, reinforcing its identification as a stage-IV salient gene in hepatocellular carcinoma (*Sarathi*

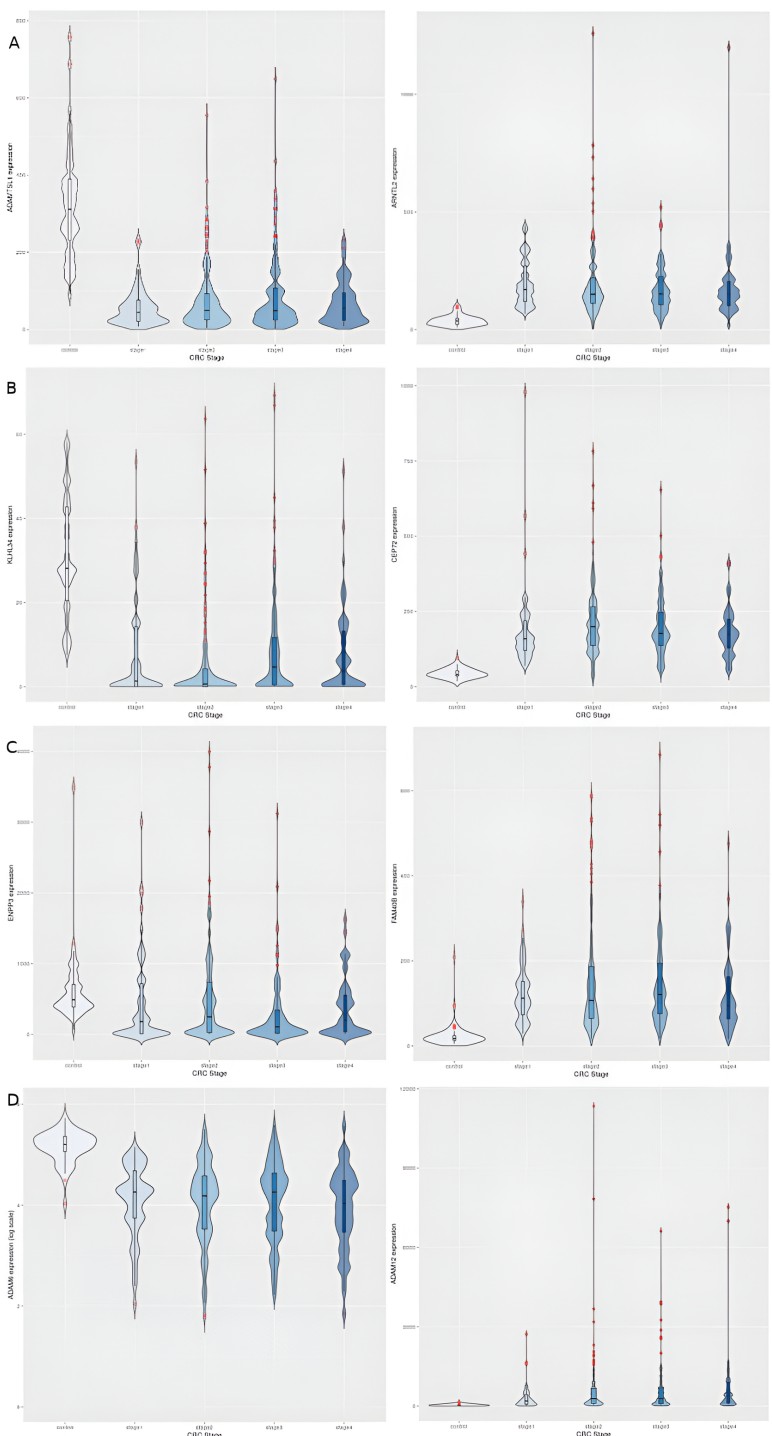

**Figure 4 Illustration of dichotomy in expression trends of stage-salient genes (namely, consistent differential upregulation and consistent differential downregulation).** Each stage is represented by one upregulated gene (column 1) and one downregulated gene (column 2). (A) Stage-I: ADAMTSL1 & ARNTL2; (B) Stage-II: KLHL34 & CEP72; (C) Stage-III: ENPP3 & FAM40B; (D) Stage-IV: ADAM6 & ADAM1. Note that the expression of ADAM6 is provided in log_10 units.

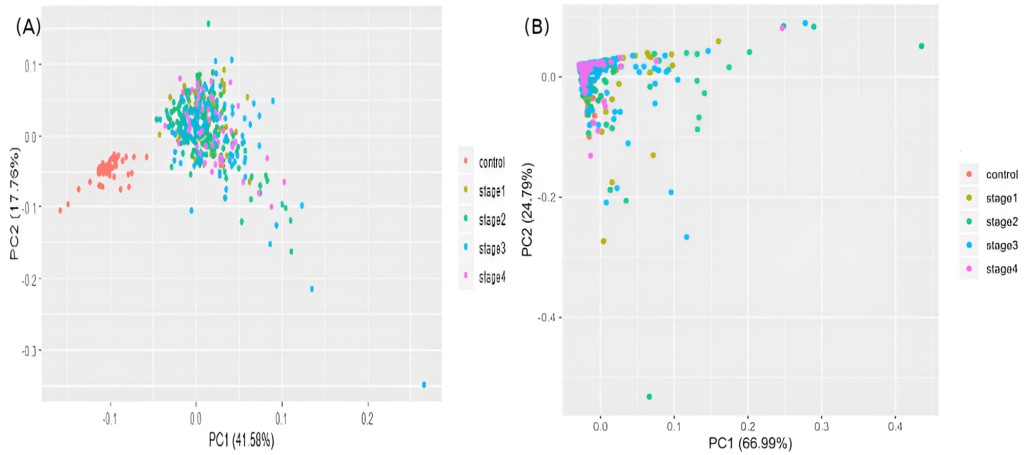

**Figure 5** **Visualizing samples in principal components space.** (A) Top 100 genes of the linear model; and (B) 100 randomly chosen genes. Only the top two principal components are used.

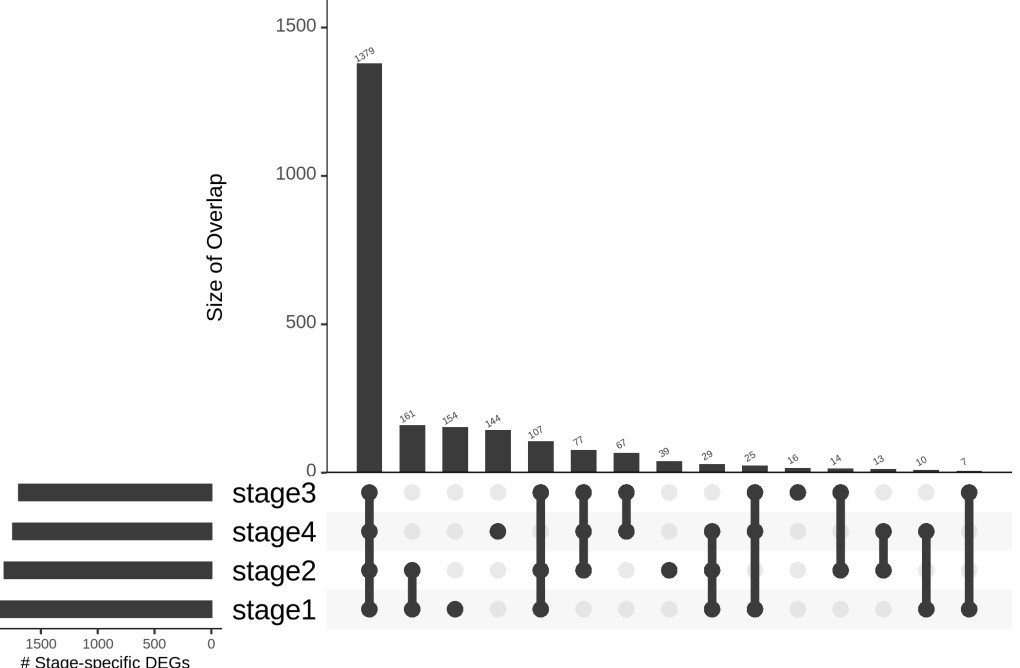

**Figure 6** **Distribution of genes based on stage-specificity.** Of the 2,242 DEGs, 1,379 appear significant in *all* the stages. It can be clearly seen that the early-stages (stages 1 and 2) share fewer DEGs with the late-stages (stages 3 and 4), flagging extra factors necessary for cancer progression to metastasis.

*& Palaniappan, 2019*), and suggesting a driver role in the metastasis of gastrointestinal cancers more generally.

Visualizing the lfc expression of stage-salient genes revealed systematic progressive expression across stages (Fig. 7). The heatmap was clustered using stage-wise expression

**Table 6  Top ten stage-salient DEGs in each stage, ordered by significance.** Such genes could represent molecular evidence for the discrete progression of colorectal cancer.

| Rank | Stage 1 | Stage 2 | Stage 3 | Stage 4 |
|------|---------|---------|---------|---------|
| 1 | CALB2 | FADS6 | PIGR | UPK2 |
| 2 | TMEM59L | EEF1A2 | MLXIPL | HOTAIR |
| 3 | JPH3 | KLHL34 | TUBAL3 | LY6G6C |
| 4 | STAC2 | DMRT3 | COMP | C6orf15 |
| 5 | NKX3.2 | GREM2 | SLC26A9 | DLX3 |
| 6 | UCHL1 | CCBP2 | CES3 | CST6 |
| 7 | KCNG1 | ADCY5 | TRY6 | VGLL1 |
| 8 | CRLF1 | PLAC2 | HABP2 | GABRD |
| 9 | C5orf23 | GPC5 | NAT2 | DKK1 |
| 10 | FBXO27 | LPHN3 | HES5 | TMEM40 |

differences w.r.to controls and showed an early-stage (stages 1 & 2) *vs* late-stage (stages 3 & 4) separation, arguing for the role of progression-significant genes in driving colorectal cancer. Clustering the genes of Table 6 yielded the following observations: (i) substantial co-clustering of stage-I with stage-II, and of stage-III with stage-IV is seen; (ii) stage-I and stage-IV genes do not intermingle; (iii) DMRT3 is the only stage-II salient gene to co-cluster with stage-IV salient genes (File S4). Further, many of the stage-4 salient genes are proto-oncogenes, steadily over-expressed in the cancer phenotype unto metastasis, whereas most of the early-stage (stages 1 and 2) salient genes are tumor suppressor genes, which are differentially down-regulated in the cancer phenotype. Even though these observations are selective and sparse, it is tempting to infer the implications for the progression pathway of colorectal cancer—initially disabling the damage-control mechanisms innate to the cell and then progressively spiraling out of control.

The results of the numeric model (Eq. (3)) sorted by significance are presented in File S5. The monotonic analysis yielded 1,944 monotonically expressed genes (MEGs; 1,389 upregulated and 555 downregulated). These are factors with a constant expression trend agnostic of stage. Applying an adj. $p$-value cutoff $< 0.05$ yielded 1,058 significant MEGs (noted in File S6). Examining the overlap of these significant MEGs with stage-salient DEGs yielded 31 progression-significant driver genes (Table 7; expression visualized in File S7). As expected, most of these biomarkers (27) are stage-4 salient DEGs, and most of them (27) are also consistently upregulated, signifying persistent unchecked cellular damage progressing to metastasis. Significant MEGs that are also significant (adj.p-val $< 1E{-}5$) in the linear and numeric models (1,186 and 997 genes, respectively) are presented in File S8. Some of the top 200 genes from the linear model (by adj. $p$-value) are also significant MEGs; these 18 genes can be found in File S9. The intersection between the top 200 genes from the numeric model and the significant monotonically expressed genes yielded 39 genes (presented in File S10). A total of 36 genes were found common to the top 200 of both the linear and the numeric (ordinal) models (File S11). Three stage-salient DEGs figured in the top 200 genes from the numeric model, namely CES3, LPHN3, and WSCD1.

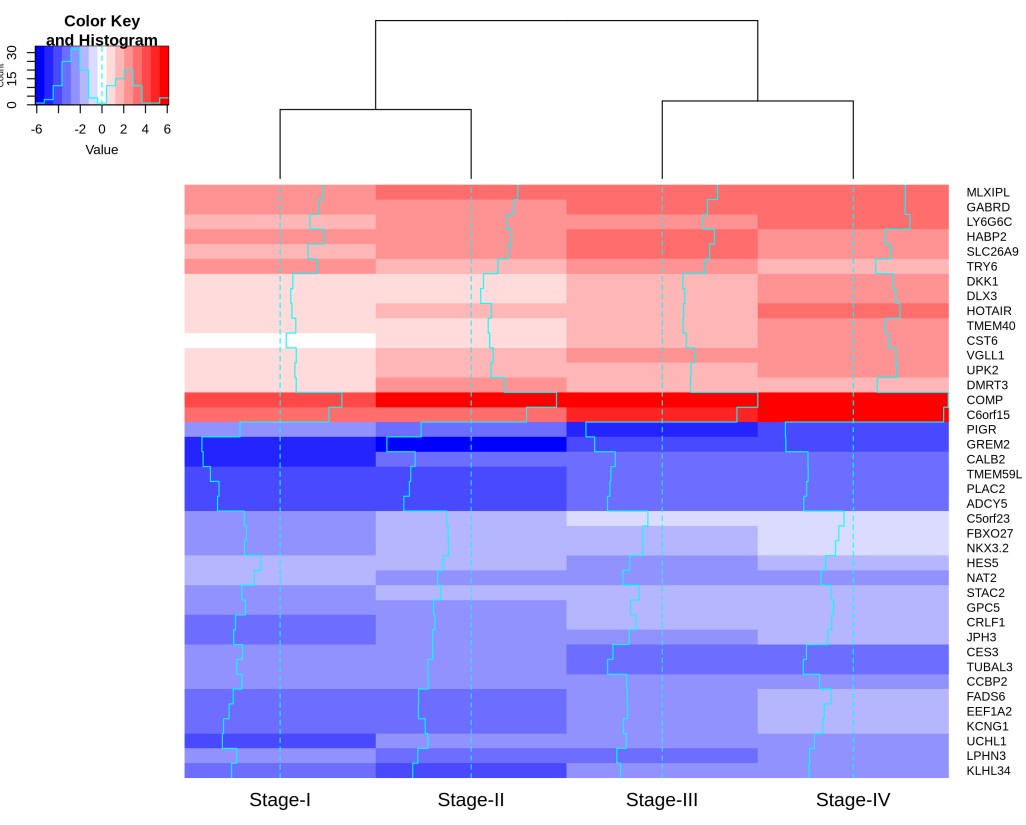

**Figure 7  Heatmap of the lfc (with respect to control samples) of top 40 genes.** Stage-salient genes express maximal salience in one of the stages. It is striking that all the ten stage-IV salient genes show monotonic progressive upregulation (for *e.g.*, GABRD). The gradient of expression is shown in the color key.

Two of the top 200 genes of the linear model were also stage-salient MEGs, namely GABRD and ESM1.

## Normals-augmented validation

To examine any negative results with the inclusion of more controls in teasing out stage-specific markers, we augmented the dataset using RNAseqDB, which added 339 normal colorectal samples. We noted that the RNAseqDB preprocessing protocol eliminated non-coding transcripts from consideration, ignoring possible expression salience of non-coding RNA biomarkers like HOTAIR. Application of our whole protocol to this controls-augmented dataset yielded a linear model, 1925 stage-specific DEGs (755 stage-I, 418 stage-II, 163 stage-III and 589 stage-IV), and 105 stage-salient markers (40 stage-I, 6 stage-II, 2 stage-III and 57 stage-IV). These are presented in File S12. We found a substantial consensus of stage-salient genes between the two datasets, with 70 biomarkers in common (Table 8; highlighted in File S12). Notably six of the top stage-I salient genes and nine of the top stage-IV salient genes were identified as salient to the respective stages with the normals-augmented dataset as well, providing robust validation for these biomarkers.

In addition, we identified a colonic cancer dataset with stage-annotation from the Gene Expression Omnibus (GEO) database (*Barret et al., 2013*), namely GSE39582, provided

**Table 7  Progression-significant driver genes, obtained by the overlap of significant MEGs with stage-salient DEGs.** A total of 31 genes sorted by the direction of fold-change (up- or downregulation) and corrected significance from the numeric model are shown. Only four genes in this group are monotonically downregulated, namely PIGR, ADH6, ATOH1, and CXCL13, while all the rest are potential proto-oncogene MEGs. It is seen that there are four stage-III salient DEGs (PIGR, DSG3, C2orf48, BIRC70) while all the rest are stage-IV salient DEGs.

| S No. | Symbol | Gene | Stage | Status | Adj.p-val |
|---|---|---|---|---|---|
| 1 | ESM1 | Endothelial cell-specific molecule 1 | IV | UP | 3.234E−16 |
| 2 | GABRD | Gamma-aminobutyric acid receptor subunit delta | IV | UP | 7.320E−11 |
| 3 | LOC283867 | Putative Long Intergenic Non-Protein Coding RNA 922 | IV | UP | 2.628E−10 |
| 4 | LY6G6E | Lymphocyte antigen 6 family member G6E | IV | UP | 1.628E−09 |
| 5 | LY6G6F | Lymphocyte antigen 6 family member G6F | IV | UP | 8.717E−09 |
| 6 | SPERT | Spermatid-associated protein | IV | UP | 3.018E−08 |
| 7 | LY6G6C | Lymphocyte antigen 6 family member G6C | IV | UP | 3.287E−07 |
| 8 | C2orf48 | Uncharacterized protein C2orf48 | III | UP | 4.499E−07 |
| 9 | TH | Tyrosine 3-monooxygenase | IV | UP | 6.419E−07 |
| 10 | NKD1 | Protein naked cuticle homolog 1 | IV | UP | 5.896E−06 |
| 11 | VGLL1 | Transcription cofactor vestigial-like protein 1 | IV | UP | 2.085E−05 |
| 12 | PLAC1 | Placenta-specific protein 1 | IV | UP | 2.822E−05 |
| 13 | COL9A3 | Collagen alpha-3(IX) chain | IV | UP | 8.310E−05 |
| 14 | SERPINE1 | Plasminogen activator inhibitor 1 | IV | UP | 1.009E−04 |
| 15 | DSG3 | Desmoglein-3 | III | UP | 1.039E−04 |
| 16 | IGFBP1 | Insulin-like growth factor-binding protein 1 | IV | UP | 5.645E−04 |
| 17 | HOTAIR | HOX antisense intergenic RNA | IV | UP | 6.808E−04 |
| 18 | ISM2 | Isthmin-2 | IV | UP | 1.377E−03 |
| 19 | LOC100133545 | C6orf15 | IV | UP | 1.471E−03 |
| 20 | DLX3 | Homeobox protein DLX-3 | IV | UP | 1.561E−03 |
| 21 | C6orf15 | Uncharacterized protein C6orf15 | IV | UP | 4.187E−03 |
| 22 | KRTAP3.1 | Keratin-associated protein 3-1 | IV | UP | 7.076E−03 |
| 23 | UPK2 | Uroplakin-2 | IV | UP | 8.241E−03 |
| 24 | C7orf52 | N-acetyltransferase 16 | IV | UP | 1.145E−02 |
| 25 | DKK1 | Dickkopf-related protein 1 | IV | UP | 1.621E−02 |
| 26 | SPDYC | Speedy protein C | IV | UP | 1.653E−02 |
| 27 | BIRC7 | Baculoviral IAP repeat-containing protein 7 | III | UP | 2.918E−02 |
| 28 | PIGR | Polymeric immunoglobulin receptor | III | DOWN | 1.226E−26 |
| 29 | ADH6 | Alcohol dehydrogenase 6 | IV | DOWN | 6.270E−15 |
| 30 | ATOH1 | Protein atonal homolog 1 | IV | DOWN | 7.378E−07 |
| 31 | CXCL13 | $C-X-C$ motif chemokine 13 | IV | DOWN | 4.675E−06 |

by the Carte d'identité des tumeurs, Ligue Nationale contre le Cancer, France (accessed 12-08-2022) (*Marisa et al., 2013*). The dataset had a large number of stage-II (271) and stage-III samples (210), relative to stage-I (38) and stage-IV (60) samples. However, only two normal samples were available, so the dataset was augmented with 308 normal colonic tissue samples from the GTEx (accessed 12-10-2022). The augmented dataset was subjected to batch correction using ComBat (*Leek et al., 2012*), and antilog$_2$ was taken to obtain the necessary counts for input to voom and the protocol described in the Methods was applied. The results are presented in File S13. Five stage-IV salient genes, namely CYP24A1, FGF19,

**Table 8  Comparison of the stage-wise salient biomarkers identified with the TCGA and the RNAse-qDB datasets.** The pval_pdt measure was applied to identify the top ten stage-2 salient and stage-3 salient genes. A substantial stage-wise consensus could be observed. The intersection of the top-10 stage-salient genes in each dataset is shown as 'Top-10 overlap.'

| Stage | No. of stage-salient biomarkers | | Size of consensus | Top-10 overlap |
|-------|------|----------|-------------------|----------------|
| | TCGA | RNAseqDB | | |
| I | 71 | 40 | 25 | CALB2, STAC2, UCHL1, KCNG1 |
| II | 10 | 10 | 5 | KLHL34, LPHN3 |
| III | 17 | 10 | 7 | HABP2, SLC26A9 |
| IV | 59 | 57 | 33 | UPK2, LY6G6C, C6orf15, DLX3, CST6, VGLL1 |

NKD1, COL9A3, and EDNRA are common to both the analyses. In addition, six stage-I salient genes, namely CPXM2, NPR3, PALM, PRDM6, TAGLN, and TPM2 are identified as stage-IV salient here. However the concordance between the markers from the reference TCGA dataset and GSE39582 is not extensive, and merits discussion. Foremost, GSE39582 is limited to colon cancer samples, which might differ in some features from rectal cancers, thereby missing some variation that is captured in the TCGA COADREAD dataset. Second, we would like to note that out-of-domain cohorts might be sensitive to distribution shifts in gene expression, which require measurement calibration with an adequate number of normals from the same (new) cohort. Since there were few normal samples in the original GSE39582 dataset, this might significantly skew the extension of gene signatures established with the reference TCGA cohort. The addition of 308 normal colonic samples available in the GTEx does not mitigate this issue, since (i) these are from an entirely different cohort, and (ii) normal rectal tissue samples remain unaccounted for. In addition, the applicability of candidate biomarker signatures to new cohorts might be bounded by bioinformatic problems pertaining to data curation and processing. The contrarian findings prompted us to seek robust validation of the models developed below.

## Development of a diagnostic aid for colorectal cancer screening

We combined the 157 stage-salient genes, top ten genes from linear modeling, and the 18 genes that were both linear and monotonically expressed into a single expression feature-space of 185 genes. The TCGA dataset was randomly split into a train dataset of 287 cancer and 41 normal samples, and a holdout testset of 71 cancer and 10 normal samples. Application of the feature selection techniques yielded a consensus feature space of just seven essential features, viz. four of the top ten linear modelling genes (ESM1, DHRS7C, OTOP3, AADACL2), two stage-salient genes (stage-2 salient LPHN3 and stage-4 salient GABRD) and one linearly monotonic gene (LPAR1). Using these features, four different ML models were trained, and hyperparameters optimized. The models were ranked on their performance on the training and holdout test sets (Table 9), and the Random Forest and 2-layer neural network models were identified for blind external validation.

**Table 9** **A summary of the models used for building a classifier capable of discriminating between cancer and normal samples based on the expression of seven features: ESM1, DHRS7C, OTOP3, AADACL2, LPHN3, GABRD, and LPAR1.** Performance in terms of balanced accuracy (average of the accuracy on either class) is reported. All models achieved 'perfection' on the holdout testset, with marginal performance variation on the training set.

| S.No | Classifier | Hyperparameters of interest | Optimal hyperparameters | Performance (bal. acc.) | |
|---|---|---|---|---|---|
| | | | | Training | Testing |
| 1 | SVM (radial kernel) | cost, gamma | 0.5, 0.1 | 99.97 | 100 |
| 2 | Random Forest | ntree (#trees in the forest), mtry (#candidate variables randomly sampled for splitting) | 500, 2.83 | 100 | 100 |
| 3 | Neural Networks (1-layer) | size, decay | 1, 1 | 99.97 | 100 |
| 4 | Neural Networks (2-layer) | #units in hidden layer 1, #units in hidden layer 2 | 4,1 | 100 | 100 |

**Table 10** **Blind evaluation of the best-performing ML models on external independent datasets.** Random Forest model was clearly better than the Neural Network 2-layer model on the external validation. Bal. acc. refers to balanced accuracy (average of sensitivity (recall) and specificity).

| S.No | Model | Bal. acc. | Specificity | Precision | Recall | F1-score | MCC |
|---|---|---|---|---|---|---|---|
| 1 | Random forest | 98.27 | 96.43 | 91.13 | 100 | 95.36 | 93.74 |
| 2 | Neural network (2layer) | 96.15 | 93.18 | 84.21 | 99.12 | 91.06 | 87.98 |

Two external datasets were chosen for blind validation: (i) Rectal_cancer_MSK (*Chatila et al., 2022*) with 113 mRNA-Seq expression samples, obtained from 738 primary rectal tumors (https://www.cbioportal.org/; accessed 02-09-2023; accessed 02-09-2023) and (ii) 308 normal colon samples from the GTEx (accessed 12-10-2022). It is noted that the microarray-based GEO datasets benchmarked in our study, namely GSE25071, GSE21510, and GSE39582 were limited in the coverage of the gene-space, lacking expression values for some of the seven features used in the models, and not further considered. The hyperparameter-optimized Random Forest and 2-layer neural network models were re-built on the full TCGA dataset and evaluated on the external datasets (Table 10). All the cancer samples were correctly predicted by the Random Forest model, yielding 'perfect' recall. There were just eleven misclassified instances out of the 421 samples in the combined external dataset, and all such instances were normal colon tissue samples, leading to a balanced accuracy of 98.27%. The Random Forest model outperformed the 2-layer Neural network model on all the metrics considered, including sensitivity, specificity, F1-score, and Matthews correlation coefficient (MCC).

## Development of a prognostic model for colorectal cancer

All the 157 stage-salient genes were subjected to univariate Cox regression analysis, and the significant biomarkers ($P < 0.05$) are presented in File S14. Of the top stage-salient genes, five emerged significant, namely JPH3, HOTAIR, CST6, GABRD, and DKK1 (all $P < 0.03$). HOTAIR, CST6, GABRD, and DKK1 are stage-IV salient, while JPH3 is stage-I salient (Fig. 8). Multivariate Cox regression analysis with feature selection yielded an

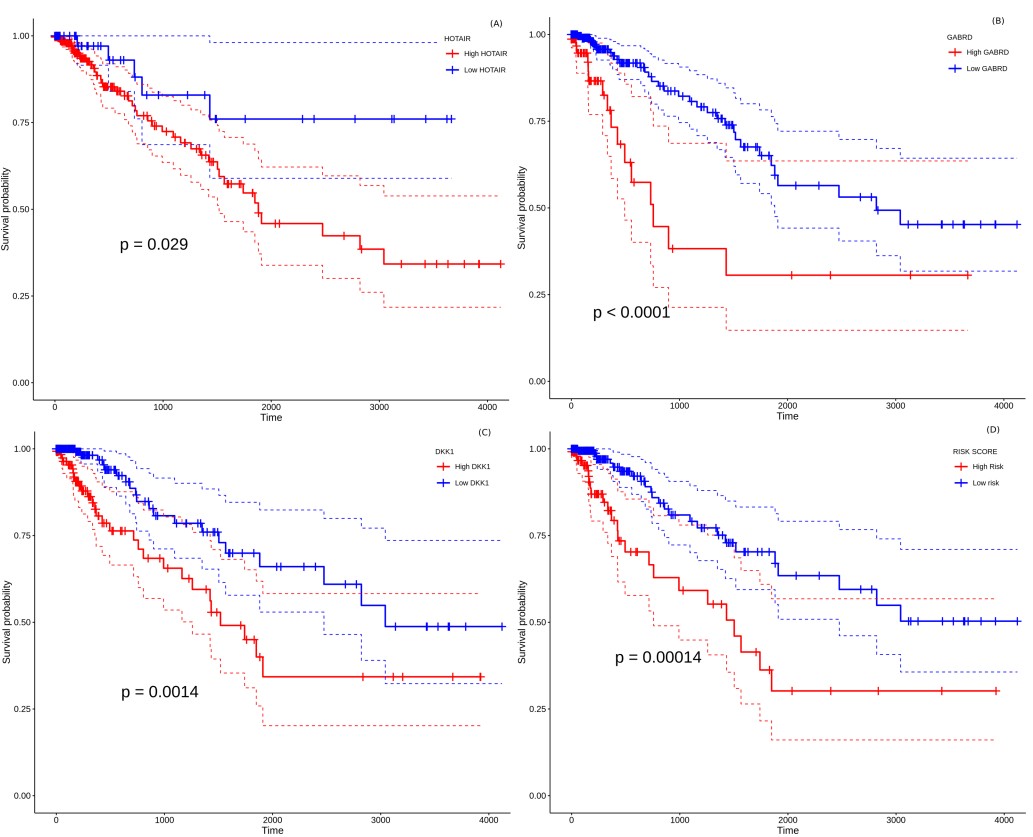

**Figure 8 Survival analysis of prognostically significant stage-salient genes.** Univariate Cox regression analysis of (A) HOTAIR , (B) GABRD, (C) DKK1; and (D) construction of optimal multivariate panel comprising the above biomarkers. Over-expression of the prognostic biomarkers has a significant effect on the survival probabilities ($P < 0.05$), and elevates the patient risk. Red - high-risk, blue - low-risk; colored dashed lines represent corresponding 95% confidence intervals.

optimal panel of three genes, namely HOTAIR, GABRD, and DKK1, with a model $p$-value $\sim$5e−04, and individual significances $\sim$0.0086, 0.0053, and 0.0238, respectively (*i.e.*, all $p$-values $< 0.05$). The multivariate risk model was given by:

Risk-score = 0.14872 * HOTAIR + 0.4423 * GABRD + 0.10877 * DKK1

The hazard rate for all the prognostic factors significantly exceeded 1.0, indicating that the constituents of the biomarker panel elevated the prognostic risk, suggesting possible oncogenic roles in line with their overexpression. The distribution of risk scores yielded a median maxstat value of 2.74 for patient risk stratification. Further, the Kaplan–Meier curve of the multivariate model suggested that the high-risk group was significantly associated ($p$-value $<0.0014$) with a poorer overall survival than the low-risk group (Fig. 12D). The model yielded an acceptable Concordance index (C-index) $\sim$0.71 $\pm$ 0.05, suggesting further application as a novel prognostic panel (*Svoboda et al., 2014*; *Niu et al., 2020*; *Sui et al., 2019*). It is significant (and perhaps not surprising) that the identified prognostic panel is entirely composed of stage-IV salient biomarkers, suggesting that the distance to

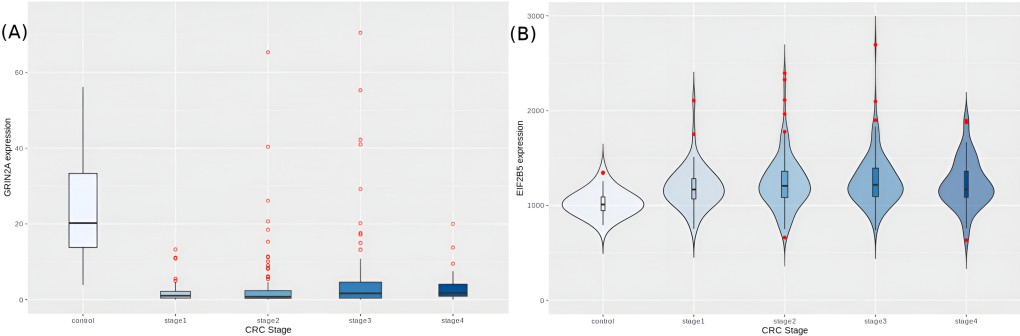

**Figure 9** **Expression trends of candidate hub-driver genes.** (A) GRIN2A and (B) EIF2B5.

metastasis is the single dominant factor in the stratification and determination of prognosis of colorectal cancer.

## DISCUSSION

To clarify the sum of findings from our studies, we began by looking at the canonical CRC drivers, APC and MSH2, which are both implicated in familial CRC. APC and MSH2 are both significantly differentially expressed (adj.p-values ~7.35e−13 and 2.06e−18 respectively). The expression patterns of these two genes (File S15) showed that APC was downregulated in the cancer phenotype, flagging its key role as a known TSG.

We then looked at the hub-driver genes identified in a previous study of CRC network analyses (*Palaniappan, Ramar & Ramalingam, 2016*), and found that GRIN2A and EIF2B5 were significantly differentially expressed in the cancer samples (adj.p-values ~2.14e−37 and 2.32e−13, respectively). GRIN2A is a TSG with least expression in stage 2 (Fig. 9A), reinforcing its role as a hub driver gene for stage 2 progression. EIF2B5 is an oncogene with maximal expression in stage 3 (Fig. 9B), again according with its identified role as a major hub driver gene for progression to advanced stages of colorectal cancer.

An analysis of the top genes from our linear model uncovered certain interesting observations. The top gene hit, CDH3 (Cadherin 3 or P-Cadherin), has been found to be overexpressed in a great majority of pancreatic ductal adenocarcinomas (PDACs) (*Taniuchi et al., 2005*), lending support to its key role in gastrointestinal cancers. Further, hypomethylation of the CDH3 promoter has been found in addition to (and the cause of) increased expression of CDH3 in both breast cancer (*Paredes et al., 2005*) and advanced colorectal cancer (*Hibi et al., 2009*). This can be due to the fact that over-expression of P cadherin leads to high motility of cells, which enables the cancer cells to metastasize.

There is emerging evidence for the role of KRT80 in head and neck squamous cell carcinoma (*Zhao et al., 2022*), but it is not a known cancer driver (https://www.intogen.org/search?gene=KRT80). The gene OTOP2 has been identified as a TSG, as it was significantly downregulated in the cancer phenotype. Another independent study also found that wild type p53 regulated OTOP2 transcription in cells, and increased levels of OTOP2 suppressed tumorigenesis *in vitro* (*Qu et al., 2019*). OTOP3 belongs to the same family of

otopetrin proton channels, but there is no published evidence for its role in any cancer (https://www.intogen.org/search?gene=OTOP3).

AADACL2 is not a known cancer driver, but there is evidence for its role in a comorbid breast-colorectal cancer phenotype (*Pande et al., 2018*). ETV4, another top candidate in our linear model, has shown significant promise as a therapeutic target. A previous study found that ETV4 knockdown in metastatic murine prostate cancer cells abrogates the metastatic phenotype but does not affect tumor size (*Aytes et al., 2013*). According to our model, ETV4 shows maximal expression in stage 4 and is concordant with a molecular basis for cancer stages. ETV4 is also a designated cancer gene in the COSMIC census (*Forbes et al., 2016*).

ESM1 was found to be clearly overexpressed in clear cell renal cell carcinoma (*Leroy et al., 2010*), and is also one among the 59 stage-4 salient genes from our study. Moreover, ESM1 is also an MEG identified in our study, placing it as a very significant driver of CRC progression. DHRS7C has been recently implicated in signaling pathways involved in glucose metabolism (*Ruiz et al., 2018*). It exerts its effects *via* mTORC2, a complex known to be at the heart of metabolic reprogramming (*Masui, Cavenee & Mischel, 2014*). Mysteriously DHRS7C was seen downregulated in colorectal cancer, given that its upregulation is necessary for glucose uptake. These observations merit experimental investigations to ascertain the precise nature of the molecular biology in question.

Some studies reveal that the LIM-domain-containing JUB serves as an oncogene in CRC by promoting the epithelial-mesenchymal transition (EMT), a critical process in the metastatic transition (*Liang et al., 2014*). The gene MTHFD1L coding for methylenetetrahydrofolate dehydrogenase 1–like is significantly overexpressed in the colorectal cancer phenotype. Studies show that MTHFD1L contributes to the production and accumulation of NADPH to levels that are sufficient to combat oxidative stress in cancer cells. The elevation of oxidative stress through MTHFD1L knockdown or the use of methotrexate, an antifolate drug, sensitizes cancer cells to sorafenib, a targeted therapy for hepatocellular carcinoma (*Lee et al., 2017*).

Comparing the transcriptomic stage-specific patterns of colorectal cancer samples identified here with their methylomic stage-specific patterns (*Muthamilselvan, Raghavendran & Palaniappan, 2022*), we uncovered interesting connections. Some of the stage-salient genes here were also identified as stage-specific differentially methylated genes, namely: BAI3, TPM2, ZSCAN18, ZNF415 (Stage-I); PLAC2, DMRT3 (Stage-II); PIGR, TUBAL3 (Stage-III); and CST6 (Stage-IV). GABRD was earlier found to be significantly differentially methylated in all stages except stage-IV, suggesting that methylation precedes the stage-4 salient change in gene expression observed in this study. In the other direction, GPX3, identified as a stage-I salient gene here, was detected as differentially methylated in stage-2, suggesting the interpretation that change in its expression is necessary for cancer metastasis and mesenchymal transition. The details for the above analysis are presented in File S16.

### Stage-1 salient DEGs

The genes CALB2 and TMEM59L cluster together in Fig. 7 with the least stage-I expression, suggesting the hypothesis that they function as tumor suppressor genes. This is supported in the literature, specifically that cells in which CALB2 is silenced do not respond to 5-flourouracil, a favored treatment for CRC, indicating that CALB2 expression is necessary for 5-flourouracil induced apoptosis (*Stevenson et al., 2011*). Another study found that heterozygosity in SNP513 of Intron 9 of the gene CALB2 might be a predictive marker for CRC (*Vonlanthen et al., 2007*). It has also been noted that increased TMEM59L expression was a pro-apoptotic indicator of cell death during oxidative stress in neuronal cells (*Zheng et al., 2017*). Regarding SOX2 and SOX10, it is noteworthy that the Cancer Genome Atlas Network observed SOX9 as a novel gene with significant recurrent mutations in COADREAD (*The Cancer Genome Atlas Network, 2012*).

### Stage-2 salient DEGs

KLHL34 was found to be hypermethylated in Locally Advanced Rectal Cancer, and knockdown of KLHL34 lowered colony formation, increased cytotoxicity, and increased radiation induced caspase 3 activity in LoVo cells (*Ha et al., 2015*). CCBP2, encoding the Chemokine decoy receptor D6, has an inhibitory effect on breast cancer malignancies due to its action to sequester pro-malignant chemokines (*Yang et al., 2013*). The lncRNA PLAC2 induces cell cycle arrest in glioma by binding to Ribosomal Protein RP L36 in a mechanism involving STAT1 (*Hu et al., 2018*). GPC5 was found to be overexpressed in the lung cancer phenotype (*Li & Yang, 2011*), in lymphoma, and in gastric cancer.

### Stage-3 salient DEGs

Copy number polymorphisms of TRY6 gene have been found in breast cancer (*Wagner et al., 2007*). HABP2 gene overexpression has been observed in lung adenocarcinoma and has been proposed as a novel biomarker for the same (*Wang et al., 2002*).

### Stage-4 salient DEGs

The lncRNA HOTAIR was found to be significantly overexpressed in HCC, and a potential biomarker for lymph node metastasis in HCC (*Geng et al., 2011*), and later implicated in different cancers (*Hajjari & Salavaty, 2015*). Another widely-cited study (*Gupta et al., 2010*) showed that enforced HOTAIR gene expression in epithelial cancer cells induces chromatin reprogramming and an increased metastatic state, while inhibition of HOTAIR inhibits cancer invasiveness. These accounts of the role of HOTAIR in metastasis accord with our findings that HOTAIR is a stage-4 salient significantly monotonically expressed biomarker. GWAS analysis identified a strong association of C6orf15 with occurrence of follicular lymphoma (*Skibola et al., 2009*). Promoter methylation of cell free DNA of the CST6 gene was found to be a potential plasma biomarker for Breast Cancer (*Chimonidou et al., 2013*). Expression of VGLL1 and its intronic miRNA miR-934 are associated with sporadic and BRCA1-associated triple negative basal-like breast carcinomas (*Castilla et al., 2014*). Expression of DKK1, an inhibitor of osteoblast differentiation, was found to be associated with the presence of bone lesions in patients with multiple myeloma (*Tian et al., 2003*). TMEM40 has been found to be a potential biomarker in patients with Bladder

cancer, serving as an oncogene and a possible therapeutic target (*Zhang et al., 2018*). The emergence of the C,E, and F members of the Lymphocyte Antigen 6 (LY6) family (*Loughner et al., 2016*; *Upadhyay, 2019*) as monotonically expressed proto-oncogenes holds promise for immunotherapy. There is a substantial evidence base for GABRD (*Gross, Kreisberg & Ideker, 2015*), which is a key component of both the screening and prognostic models developed here. Consistent expression trends in GABRD and other stage-salient MEG DEGs provide unmistakable evidence for the existence of molecular signatures in CRC progression.

## Benchmarking with curated databases

We found 13 of the top 200 genes from the linear model documented in the CGC v84 as known cancer genes (Table 11). Two genes, MACC1 and SALL4, were specifically documented for colorectal cancer. HSP90AB1 had been earlier identified as a top MEG in HCC (*Sarathi & Palaniappan, 2019*). Screening the 157 stage-salient genes against the NCG7.0, which is a curated database of cancer drivers and healthy drivers, yielded 28 genes, of which eight were in the top 40 stage-salient genes (File S17). All the hits were documented to carry mutations in their coding region (*vs* noncoding region). Three were *canonical* oncogene drivers, namely HOXC11, SOX2, and KCNJ5, while the rest 25 are putative oncogenes and putative tumor suppressors in almost equal measure. Two stage-salient genes, namely CNTN1 and BAI3 (ADGRB3) were documented as putative tumor suppressor genes involved in gastric adenocarcinoma, providing specific support for our findings. PIGR is identified as an essential healthy driver (*Olafsson et al., 2020*), signifying that mutations in this gene confer an exceptional protective effect, and its down-regulation could drive tumorigenic processes. Intriguingly, the stage-salient genes C5ORF23 (NPR3), SOX2, and KCNJ5 are the only instances where the documentation is dissonant with our primary findings; these three were marked as putative oncogenes, though they are identified as down-regulated here. Further investigations in this direction are warranted to set the literature straight. Documentary evidence for drugs targeting any of these genes is absent, emphasizing the value of the present study in pinpointing novel candidates for diagnosis, therapy and prognosis. To perform a systematic analysis of therapeutic interventions based on these targets, we consulted ClinicalTrials.gov for clinical trials targeting stage-salient genes. Ten genes from the top stage-salient genes are being pursued in clinical trials, either as target or endpoint, colorectal or other cancers. Details of clinical trials along with the current status/phase of each trial are provided in File S18. DKK1 and HOTAIR are the only stage-4 salient genes implicated as targets/endpoints in clinical trials. DKK1 is involved in three clinical trials for colorectal and gastric cancers. HOTAIR is the target of a clinical trial for thyroid cancer (ctgov:NCT03469544) (*Abudoureyimu et al., 2016*). HOTAIR is documented in the NONCODE database (http://www.noncode.org/) as disease-associated, specifically with colorectal cancer (ID: NONHSAG011264.3), validating its role in oncogenic processes. It is notable that GABRD is not a target in any of the registered clinical trials, flagging a prime potential interest for future efforts. LPHN3, a stage-2 salient gene, is targeted in four clinical trials aimed against metastatic colorectal cancer, to explore possible therapeutic efficacy in thwarting cancer progression prior to

**Table 11 Summary of top 200 genes of the linear model documented in the CGC.** These are cancer driver genes with known experimental evidence. In the case of FAM135B, FEV, CBFB, and CTNND2, the regulatory status inferred here is at odds with their documented cancer role, thereby indicating potential anomalous regulation whose resolution would be tractable to experimental investigation.

| Gene symbol | Illustrative tumors | Documented role | Status |
|---|---|---|---|
| ETV4 | Ewing sarcoma, prostate carcinoma | oncogene, fusion | UP |
| CBFB | acute myeloid leukemia | TSG, fusion | UP |
| KIAA1549 | pilocytic astrocytoma | fusion | UP |
| HSP90AB1 | non-Hodgkin's lymphoma | fusion | UP |
| MACC1 | hepatocellular carcinoma, *CRC* | oncogene | UP |
| SET | T-cell acute lymphoblastic leukemia | oncogene, fusion | UP |
| MET | papillary renal, head-neck squamous cell | oncogene | UP |
| SALL4 | *CRC*, breast cancer, prostate cancer, glioblastoma, melanoma | oncogene | UP |
| FAM135B | small cell lung cancer, esophageal cancer | oncogene | DOWN |
| FEV | Ewing sarcoma | oncogene, fusion | DOWN |
| CDH10 | Melanoma | TSG | DOWN |
| PHOX2B | Neuroblastoma | TSG | DOWN |
| CTNND2 | prostate adenocarcinoma, GIST (gastrointestinal stromal tumor) | oncogene | DOWN |

irreversible outcomes. FADS6 (a stage-II salient gene) is an endpoint in a clinical trial to treat colorectal adenomatous polyps, which is a precursor to malignant lesions. CALB2 and C5orf23 (NPR3) are each involved in one clinical trial related to colorectal cancer. Some stage-salient genes are being pursued in treatment of cancers in other cell types/tissues, underlining the role played by certain genes in contributing to general cancer hallmark processes (*Hanahan & Weinberg, 2011*). Specifically NAT2 is a target in nine different clinical trials against diverse cancers, significantly highlighting its essential role in driving hallmark processes in unrelated cancers.

## Insights from network analysis

Stage-wise network analysis of colorectal cancer progression has shed light on certain genes potentially underlying progression (*Rahiminejad et al., 2022*). The strength of the computational evidence for the candidate biomarkers identified herein urged a network analysis to examine the findings in a larger context. The intersection between the sets of all stage-salient biomarkers and the significant MEGs might highlight monotonically enriched pathways essential to the pathophysiology of colorectal cancers. Hence the 31 stage-salient MEGs were chosen to reconstruct the STRING network, with 50 interactors in the first shell and 10 interactors in the second shell. This yielded a PPI with 235 edges with an extremely significant enrichment $p$-value $< 1.0e-16$ (Fig. 10). A Gene Ontology (*Ashburner et al., 2000*) analysis of this reconstructed network showed enrichment for the Wnt-Frizzled-LRP5/6 complex component at $p$-value $< 1E-04$. An analysis with KEGG (*Kanehisa et al., 2016*) showed enrichment for 2-oxocarboxylic acid metabolism at $p$-value $\sim 0.001$, indicating a Warburg-shift in metabolism. An analysis with Reactome (*Sidiropoulos et al., 2017*) showed significant enrichment of SMAD2/3 and SMAD4 MH2

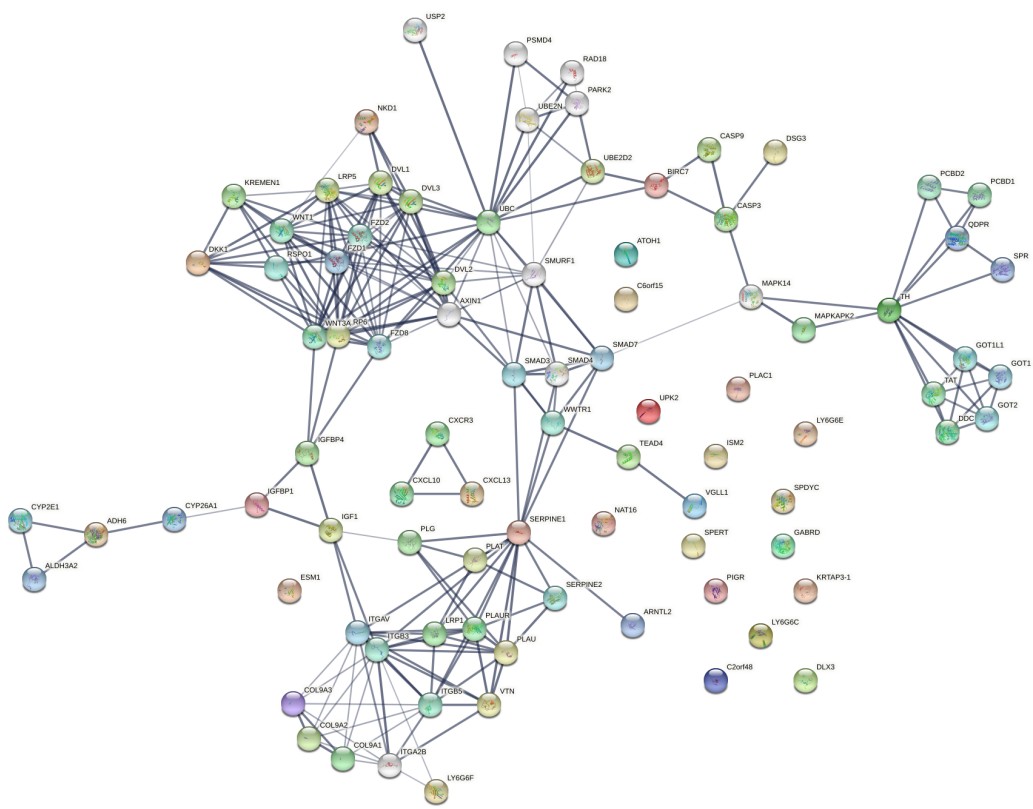

**Figure 10  Network reconstruction of perturbed pathways with monotonic expression enrichment based on the seed set of stage-salient MEGs in TCGA COADREAD.** Evidence from known interactions (curated databases, experimentally determined) or predicted from gene neighborhood, gene fusions or gene co-occurrence were used in identifying edges. Colored nodes indicate query proteins and first shell of interactors, whereas white nodes indicate second shell of interactors.

Domain Mutants in Cancer (*p*-value < 0.01). These observations *in toto* provide striking evidence for the involvement of these biomarkers in driving CRC progression.

Isolated nodes in the network included GABRD, DLX3, ISM2, LY6G6C & E, SPYDC, UPK2, C2orf48, PIGR, KRTAP3-1, C7orf52 (NAT16), SPERT, and PLAC1. All the isolated nodes are proto-oncogenic (see Table 7), hence could provide targets for inhibition in personalized cancer medicine. An outlier component (not in the giant connected component) was made of the CXCL chemokine family, stemming from CXCL13—a recently recognized immune checkpoint with a key role in tumor progression (*Yang et al., 2021*; *Ren et al., 2022*). This component could constitute a novel target for upregulation in CRC immunotherapy. A drug-repurposing search with the DrugGeneBadger (*Wang et al., 2019*) for each of the 31 stage-salient MEGs yielded drugs (small molecules with q-values < 0.05) to pharmacologically alter the expression of these identified biomarkers. The search revealed that curcumin is effective against at least 13 of these targets, and piperlongumine against eight of these targets. Six biomarkers (HOTAIR, ISM2, KRTAP3-1, SPDYC, LY6G6F, and NKD1) found no drug available in LINCS1000 (*Subramanian et al., 2017*) to
modulate their expression, and these constitute potential novel targets for drug discovery against metastatic transition in CRC.

A network specific to colon cancer could be obtained using the results for GSE39582, and is presented in File S19. Enrichment analysis of this network with Gene Ontology indicated significance for Arp2/3 complex-mediated actin nucleation ($p$-value $\sim$1e−4), which is known to contribute to invasive colorectal cancer (*Zheng et al., 2023*). A KEGG analysis showed enrichment for oxidative phosphorylation ($p$-value $\sim$1e−20), with a prominent clustering of NDUF and COX gene families. A Reactome analysis showed a minor enrichment of enzymatic protein conjugation processes (UBE2I, UBA2, SAE1) that monitor intracellular proteins and cell states ($p$-value $\sim$0.02). These findings indicate an enrichment of proliferation-independent metabolism-rewiring pathways necessary for colorectal cancer progression, and could be contrasted with the analyses in *Marisa et al. (2013)*.

## Immune-cell infiltration analysis

Deconvolution of the TCGA samples based on the LM22 immno-cyte signature with 100 permutations yielded 107 samples with significance ($P < 0.05$), including eleven controls. These samples, with their TCGA identifiers, are presented in File S20. The significant samples were analyzed for the relative abundance of the 22 immune cell types. A heatmap of the sample-wise immune cell-type proportions was generated (File S21A), and the clustering patterns of the cell-types across samples was visualized using a dendrogram. We observed the following clusters: mast cells resting and plasma cells; mast cells activated and neutrophils; T cells CD8, T cells follicular helper, and macrophages M1; T cells CD4 memory resting and B cells naive. The macrophages M0 and M2 were clear outgroups in the dendrogram. A normalized stacked bar chart of the sample-wise immnuo-cyte fractions revealed substantial variations in immune cell-type composition between normal and cancer samples (File S21B). To investigate further, we analyzed the differences in distribution of cell proportions between normal and tumor samples for each immune cell-type (File S21C; data presented in File S20). Eight of the 22 immunocyte types showed significant distribution differences (adj. $P < 0.05$). Specifically, we found the infiltration of four immune cell-types preferentially enriched in tumor samples, namely macrophages M0, T cells CD4 memory activated, mast cells activated, and neutrophils, while four other immune cell-types were preferentially depleted in tumor samples, namely macrophages M2, T cells CD4 memory resting, mast cells resting, and plasma cells. In particular, macrophages M0 exhibited both the largest effect size ($>2.0$) and the greatest significance ($<1E-07$) of infiltration in tumor samples. The preferential enrichment of mast cells activated and T cells CD4 memory activated *versus* the preferential depletion of mast cells resting and T cells CD4 resting suggested that tumorigenesis activates resting immune cell-types, potentiating their infiltration of the tumor microenvironment. To integrate these observations, we computed the correlation matrix of the immune cell-types based on their sample-wise proportions over both normal and tumor samples (File S21D). The largest positive correlations were exhibited by T cells follicular helper with T cells CD8 (Pearson's $\rho \sim$0.52), and with macrophages M1 (Pearson's $\rho \sim$0.45), reinforcing their clustering in

the dendrogram. Intriguingly, the largest negative correlation (in magnitude) was exhibited by macrophages M0 and T cells CD4 memory resting (Pearson's $\rho \sim -0.51$) (File S21D). Given that macrophages M0 are preferentially enriched in tumor samples whereas T cells CD4 resting and mast cells resting (Pearson's $\rho \sim -0.47$ with macrophages M0) are both preferentially depleted, these observations cohere and could hold preliminary significance for immunotherapy. Discovery of multicellular immunocyte community structures could pave the way for personalized immunotherapy in CRC treatment (*Sarathi & Palaniappan, 2019*; *Ge et al., 2019*).

## COADREADx

Based on the external validation, the Random Forest model was identified as the best model for screening early-stage cancer. Coupled with the prognostic model, these could aid the risk stratification of patient samples. With this application in mind, we have deployed COADREADx, an experimental web service for the screening of patient samples as 'cancer' or 'normal', and subsequent prognostication in the case of 'cancer'. COADREADx has been implemented using R-Shiny (https://shiny.rstudio.com/), and is available for academic use at: https://apalanialab.shinyapps.io/coadreadx/. A help document with sample input files for different use-cases, and a companion how-to video have been made available on the landing page. To aid the effective interpretation of COADREADx predictions, the prediction probability for the predicted diagnostic class is provided, yielding a level of confidence in the prediction. Similarly the risk stratification of 'cancer' samples is accompanied by the quantile of the estimated risk-score as well as its fold-change from the median value of the risk score distribution. These values suggest the strength of evidence for the predicted risk class.

In summary, we have performed a novel *de novo* analysis of the TCGA COADREAD gene expression dataset, and identified multiple interesting classes of biomarkers. The biomarkers have been validated with alternative datasets, network analysis and immune cell infiltration analysis. Some of the biomarkers could suggest novel hypotheses for targeted therapy and immunotherapy. Using purifying techniques, we have carved feature spaces from these biomarkers to build screening and prognostic models of colorectal cancer. The screening model has been externally validated, while the prognostic model has been bootstrapped for confidence. Both the models have been deployed *via* COADREADx, a web-server designed to return confidence estimates for all its predictions. Phenomena of distribution drift and shift in new samples and out-of-domain cohorts challenge the applicability of COADREADx, which might need refinement in the light of such evidence. Further validation of the models on colon cancer samples might also be warranted. Enabling risk stratification is vital to treatment strategy and clinical management of the cancer. Thus experimental validation and further improvement of COADREADx is necessary to demonstrate its clinical utility for screening and prognosis purposes. It is reckoned that the availability of such software-as-medical-devices could ease the accessibility to effective surveillance technologies for early detection of colorectal cancer (*Muthamilselvan, Ramasami Sundhar Baabu & Palaniappan, 2023*).

## CONCLUSIONS

We have identified stage-salient signatures of colorectal cancer, and developed multiple workflows toward their computational validation. Early-stage biomarkers present prime targets for potential pharmacological intervention, while modulating the expression of progression-significant biomarkers (for *e.g.*, by inhibiting the overexpressed ones or activating the expression of downregulated ones) could possibly block the progression of colorectal cancer. A model for the early-stage screening of colorectal cancer was created using seven consensus biomarkers (namely ESM1, DHRS7C, OTOP3, AADACL2, LPHN3, GABRD, and LPAR1), and yielded >98% balanced accuracy on external validation. A survival analysis protocol yielded a prognostic panel of three stage-IV salient genes (namely HOTAIR, GABRD, and DKK1) for patient risk stratification, suggesting that high-risk prognosis could be extracted from the oncogenic signature of these metastasis-salient genes. The weight of the evidence presented herein suggests a central role for molecular factors in cancer progression. COADREADx, provides an experimental set of tools for colorectal cancer screening and prognosis based on the candidate biomarkers identified in our study. Our findings will need experimental validation and testing in prospective cohorts for translation to the clinic, and set the stage for further exploration of signature panels on the overall path to securing the best intervention for the condition. The hypothesis-agnostic study design also provides a framework for the investigation of other cancers, and more generally, progressive degenerative conditions.

## ACKNOWLEDGEMENTS

A portion of this work was performed when A.P. was a visiting faculty at the University of Wisconsin–Madison. We would like to acknowledge additional computing support from Google TPU Research Cloud. SASTRA Deemed University provided infrastructure, resources, and support.

### Funding

This work was supported by DST-SERB grant no. EMR/2017/000470 (to Ashok Palaniappan). The funders had no role in study design, data collection and analysis, decision to publish, or preparation of the manuscript.

### Grant Disclosures

The following grant information was disclosed by the authors:
DST-SERB: EMR/2017/000470.
SASTRA Deemed University provided infrastructure, resources, and support.

### Competing Interests

The authors declare there are no competing interests.

## Author Contributions

- Ashok Palaniappan conceived and designed the experiments, performed the experiments, analyzed the data, prepared figures and/or tables, authored or reviewed drafts of the article, and approved the final draft.
- Sangeetha Muthamilselvan performed the experiments, analyzed the data, prepared figures and/or tables, and approved the final draft.
- Arjun Sarathi performed the experiments, analyzed the data, prepared figures and/or tables, and approved the final draft.

## Data Availability

The data is available at Figshare: Muthamilselvan, Sangeetha; Sarathi, Arjun; Palaniappan, Ashok (2022). COADREADx: A comprehensive algorithmic dissection of colorectal cancer unravels salient biomarkers and actionable insights into its discrete progression. figshare. Online resource. https://doi.org/10.6084/m9.figshare.20489211.v5.

The code for COADREADx model development and R Shiny app is available at Zenodo: Ashok Palaniappan. (2024). APalaniaLab/COADREADx: COADREADx_models (v1.0). Zenodo. https://doi.org/10.5281/zenodo.13790220.

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
