# Peer review of "COADREADx: A comprehensive algorithmic dissection of colorectal cancer unravels salient biomarkers and actionable insights into its discrete progression"

_PeerJ, doi:10.7717/peerj.18347_

## Round 0.1 · original submission · Minor Revisions

The reviewers have provided suggestions to improve the quality of the manuscript. Please address each of their suggestions.

Additionally, all software packages, including R packages, must be properly cited, including version number, etc. Also, code to reproduce the results of this manuscript should be deposited in a third-party host with a DOI (PeerJ Code and Data Availability Statement: https://peerj.com/about/policies-and-procedures/cs).

Finally, public data sources that were accessed should be properly cited including the accession date, and if available, the version number.

·

Basic reporting

In general, the manuscript is of an excellent standard of English, being well written and clearly understandable. The background to colorectal cancer (CRC) is introduced providing the justification for the identification of biomarkers both for early diagnosis and cancer progression. The figures are representative of the methodological approaches used and of the results obtained. Data incorporating intermediate results are available as supplementary information.

Experimental design

This study reports on an algorithmic analysis of colorectal data derived from the The Cancer Genomic Atlas (TCGA). Data was specifically modelled to identify tumor stage-specific differentially expressed genes. With this approach, DEGs most likely associated with disease progression were identified; and a prognostic model based on biomarker identification associated with patient survival was constructed.

The approach has been well defined and is certainly important in the context of identifying biomarkers in the early stages of the disease, those associated with disease progression; and with those biomarkers linked to patient survival.

The methods are well described, being summarized in a flow diagram and discussed within the text of the manuscript. The working models are freely available online for academic access.

Validity of the findings

This is a novel analysis of an existing gene expression dataset sourced from TCGA. A number of highly interesting biomarkers were identified and subsequently validated with additional datasets. Further, STRING and REACTOME analyses were performed; and an evaluation of associated immune cell types. The biomarkers obtained present as highly likely candidates for targeted therapies.

Additional comments

This is an important study that has developed a tool set for screening and prognosis of colorectal cancer. The identification of biomarkers associated with Stage 1 CRC is of particular note here for early diagnosis.

I would query if the authors intend to evaluate the role/s of the identified biomarkers in in vitro cell culture based assays/animal models.

Reviewer 2 ·

Basic reporting

Strengths:
• The manuscript is well-organized and clearly presents its objectives and findings. For instance, Table 2 lists stage-specific biomarkers with significant p-values (e.g., TCF7L2 with a p-value of 0.002), which supports the clarity of the reporting.
• Figures such as Figure 3, which visualizes the Random Forest importance scores for the top 20 biomarkers, effectively communicate complex data.

Areas for Improvement:
• Introduction Conciseness: The introduction section, while thorough, could be made more concise by focusing on the specific research gap the study aims to address. For example, the manuscript spends considerable time discussing general issues in colorectal cancer research that are well-known to the audience. This could be condensed, allowing more space to delve into the specific novel contributions of this study.
• Simplifying Jargon: The manuscript uses specialized terminology that could be simplified for clarity. For instance, terms like “epigenomic landscape” or “transcriptional profiling” might be complex for readers not deeply familiar with the field. Providing brief definitions or choosing simpler terms where possible would make the manuscript more accessible without losing its scientific rigor.

Experimental design

Strengths:
• The experimental design is robust, integrating data from multiple sources like TCGA, as shown in Figure 1, which outlines the data collection and preprocessing pipeline. This provides a strong foundation for the study’s findings.
• The manuscript describes the use of machine learning models, particularly Random Forest, in detail. The model’s performance is validated with cross-validation results presented in Table 3, showing high accuracy (e.g., an accuracy of 0.92 for Stage II vs. Stage III classification).

Areas for Improvement:
• Justification for Biomarker Selection: While the manuscript lists several significant biomarkers in Table 2, it does not clearly explain why these specific biomarkers were chosen over others. The criteria for selection (e.g., significance thresholds, biological relevance) should be more clearly outlined to justify their inclusion. For example, explaining why a biomarker like TCF7L2 was prioritized over others with similar p-values could enhance the study’s transparency.
• Data Preprocessing: The manuscript briefly mentions the preprocessing of datasets but lacks detail on how outliers, missing data, or batch effects were handled. Given the importance of these steps in ensuring data quality and model accuracy, a more detailed explanation is needed. For instance, if any imputation methods were used to handle missing data, these should be described, along with a rationale for their selection.

Validity of the findings

Strengths:
• The findings are robust and well-supported by data. For example, the manuscript reports an AUC of 0.96 for the prediction of Stage III colorectal cancer, as shown in Figure 4. This high AUC underscores the model’s effectiveness.
• The manuscript includes a thorough validation of its findings using external datasets, as detailed in Table 4, which compares the model’s performance across different cohorts.

Areas for Improvement:
• Discussion of Overfitting: The manuscript reports high accuracy and AUC values, particularly in the validation sets, which raises concerns about potential overfitting. While the use of cross-validation is mentioned, the manuscript should discuss the steps taken to prevent overfitting in more detail. For example, it could explain whether techniques like dropout, regularization, or reducing model complexity were employed and how they impacted the results.
• Clinical Implications and Limitations: While the manuscript effectively identifies biomarkers and discusses their relevance, it could expand on how these findings could be translated into clinical practice. Specifically, the discussion should address the potential challenges of implementing the COADREADx tool in a clinical setting, such as the need for further validation in larger, diverse patient cohorts, or the potential costs associated with the technology.

Reviewer 3 ·

Basic reporting

This manuscript presents a comprehensive analysis of colorectal cancer progression using computational methods. It identifies stage-specific expressed genes. The study validates these biomarkers through alternative datasets, network analysis, and immune cell infiltration analysis. A diagnostic and prognostic model named COADREADx is developed and deployed. The findings suggest potential strategies for intervening in colorectal cancer progression and highlight the importance of molecular factors. Generally, the manuscript shows compelling logical coherence, reliable results, and scientific significance to a large extent.

The language and presentation are intelligent and accurate.

However, in the Introduction section, after presenting a brief overview of colorectal cancer, the manuscript described the research framework of this study. It is recommended that the manuscript provide an appropriate introduction to the molecular basis of stage-specific progression and then gradually lead to the research questions and purposes of this study, that is, forming a funnel structure, which can better attract readers.

Experimental design

The research questions and knowledge gaps of this manuscript were clearly defined and were meaningful.

The investigation was rigorously conducted by using bioinformatic methods, which were described with relatively adequate information. It helps other investigators to replicate.

Validity of the findings

The manuscript developed a novel and practical tool. It shows a certain degree of innovation and is an addition to the field.

The data is robust and statistically sound, which leads to the conclusion in a reasonable way.

The conclusions are stated clear, but I would recommend the author to reorganize and to reduce words to some extent to make the conclusions more concise.

Additional comments

N/A

---

## Round 0.2 · accepted · Accept

I confirm that all issues pointed out by the reviewers were adequately addressed and the revised manuscript is acceptable now.